# In Vitro Influence of a Chemically Characterized *Hippophae rhamnoides* L. Fruit Extract on Healthy and Constipated Human Gut Microbiota Functionality and Aquaporin-3 Expression

**DOI:** 10.3390/foods14213800

**Published:** 2025-11-06

**Authors:** Lorenza Francesca De Lellis, Ángela Toledano-Marín, Miguel Navarro-Moreno, Elisabetta Caiazzo, Gennaro Madonna, Adriana Delgado-Osorio, Daniele Giuseppe Buccato, Luana Izzo, Antonio Paolillo, Alessandro Di Minno, Hammad Ullah, Maria Vittoria Morone, Anna De Filippis, Massimiliano Galdiero, Armando Ialenti, José Ángel Rufián Henares, Maria Daglia

**Affiliations:** 1Department of Pharmacy, University of Napoli Federico II, Via D. Montesano 49, 80131 Naples, Italy; lorenzafrancesca.delellis@unina.it (L.F.D.L.); elisabetta.caiazzo@unina.it (E.C.); gennaro.madonna@unina.it (G.M.); danielegiuseppe.buccato@unina.it (D.G.B.); luana.izzo@unina.it (L.I.); antonio.paolillo@unina.it (A.P.); alessandro.diminno@unina.it (A.D.M.); armando.ialenti@unina.it (A.I.); maria.daglia@unina.it (M.D.); 2Departamento de Nutrición y Bromatología, Instituto de Nutrición y Tecnología de los Alimentos, Centro de Investigación Biomédica (CIBM), Universidad de Granada, 18071 Granada, Spain; antolemarin@correo.ugr.es (Á.T.-M.); miguelnav@ugr.es (M.N.-M.); adrianadelgado@ugr.es (A.D.-O.); jarufian@ugr.es (J.Á.R.H.); 3School of Pharmacy, University of Management and Technology, Lahore 54000, Pakistan; 4Department of Experimental Medicine, Section of Microbiology and Clinical Microbiology, University of Campania “L. Vanvitelli”, 80138 Naples, Italy; anna.defilippis@unicampania.it (A.D.F.); massimiliano.galdiero@unicampania.it (M.G.); 5International Research Center for Food Nutrition and Safety, Jiangsu University, Zhenjiang 212013, China

**Keywords:** short-chain fatty acids (SCFAs), polyphenol bio-accessibility vegetable extract, gastro-intestinal motility, constipation, intestinal transit regulation, aquaporins, gut microbiota functionality

## Abstract

To identify the underlying mechanisms by which *H. rhamnoides* fruit extract exerts regulatory effects on intestinal function, we investigated its chemical composition using UHPLC Q-Orbitrap HRMS and evaluated its biological effects on Aquaporin-3 (AQP-3) expression via Western blot in the intestinal epithelial cell line (HT-29). Moreover, fecal microbiota from healthy and constipated adults was employed to mimic the in vitro fermentation of the digested extract and evaluate its effects on gut microbiota functionality. Antioxidant capacity (i.e., Total Phenolic Contents (TPC), 2,2′-azino-bis(3-ethylbenzothiazoline-6-sulfonic acid) (ABTS), 2,2-diphenyl-1-picrylhydrazyl (DPPH), and ferric reducing antioxidant power (FRAP) assays) was assessed prior to and after simulated digestion and fermentation processes. Short-chain fatty acids (SCFAs) were quantified using UHPLC-RID of the fermented samples. In the extract, 23 compounds belonging to a variety of classes (mainly polyphenols) were tentatively identified. The extract significantly upregulated AQP-3 expression in the absence of cytotoxicity. After in vitro fermentation with gut microbiota isolated from constipated subjects, ABTS and FRAP values significantly decreased, as well as TPC, suggesting a greater consumption of antioxidant compounds, consistent with the increased production of radical compounds associated with constipation. Fermentation with intestinal microbiota with healthy and constipated gut microbiota resulted in an increase in SCFA. These results provide preliminary insights into a non-pharmacological strategy for functional constipation.

## 1. Introduction

Constipation is characterized by a pattern of symptoms, including hard or lumpy stools, abdominal discomfort, infrequent or difficult bowel movements, and/or a sensation of incomplete evacuation. Functional constipation (FC), a distinct constipation subtype, usually occurs in the absence of anatomical abnormalities or underlying systemic disease and is primarily influenced by unsuitable dietary habits, psychological stress, and altered composition and function of intestinal microbiota (dysbiosis) [1]. Based on the Rome IV criteria [2], the global prevalence rate of chronic idiopathic constipation is estimated at ≈14% [3]. Current clinical practice primarily relies on a combination of dietary and lifestyle modifications alongside the use of laxatives as a first-choice approach to manage FC, with emphasis on increasing dietary fiber intake (up to 25–30 g/day) to exert prebiotic activity, stimulating the gut microbiota to produce short-chain fatty acids (SCFAs), and helping retain water in the colon, thereby increasing stool bulk [4,5,6,7]. However, despite the widespread availability of over-the-counter laxatives and dietary fiber food supplements, most individuals with FC report incomplete symptom relief. Moreover, high dietary fiber doses cause side effects, e.g., bloating, gas, and abdominal distension, which greatly contribute to poor treatment adherence [8]. Several botanical species exert potential benefits to improve FC symptoms [9]. In parallel, the use of traditional laxative plants, such as *Senna alexandrina* Mill., *Aloe vera* (L.) *Burm.f.*, *Frangula purshiana* (DC.) *A.Gray ex J.G.Cooper*, is restricted in Europe due to safety concerns. Indeed, the EFSA (European Food Safety Authority) is currently monitoring the carcinogenic potential of hydroxyanthracene derivatives they contain [10]. This underscores the need to explore effective and safer alternatives [9].

*Hippophae rhamnoides* L., commonly known as sea buckthorn, has a long history of traditional use in both Europe and Asia for a wide range of health conditions. While due to their tannin content, *H. rhamnoides* leaf extracts are used to treat diarrhea [11], fruit extracts are used to treat jaundice, respiratory tract diseases (asthma), cardiovascular risk factors (hypercholesterolemia and hyperglycemia) and gastrointestinal disorders, including constipation [12,13,14]. This wide range of effects is linked to the fruit content of nutritional and bioactive components, e.g., carbohydrates, lipids (fatty acids and phytosterols), amino acids, dietary fiber (cellulose, hemicelluloses, pectin, lignin), fat-soluble vitamins (A, E and K), vitamin C, organic acids, polyphenols, and terpenes [15]. Recently, an *H. rhamnoides* fruit hydro-methanol extract was tested in ex vivo and in vivo model systems for its therapeutic effectiveness in constipation [16]. The results were consistent with a laxative and prokinetic effect of *H. rhamnoides* fruit extract and increased fecal production, only in part mediated by activation of muscarinic receptors. The authors’ conclusions suggested that the extract may exert its effects via uncharacterized and as-yet unexplored mechanisms of action [16].

Here, we have evaluated the metabolite profile of a hydroethanolic extract from *H. rhamnoides* fruits to improve our knowledge of its chemical composition and identify underlying mechanisms of action through which this extract exerts its regulating effect on intestinal function. The in vitro bio-accessibility following simulated oro-gastrointestinal digestion and colonic fermentation by means of intestinal microbiota isolated from both healthy and constipated subjects was also assessed. The main results indicate that, following digestion and fermentation with the microbiota isolated from constipated subjects in *H. rhamnoides* extract, there is greater consumption of compounds with antioxidant activity, in line with the greater production of radical compounds, probably induced by constipation. Furthermore, the extract globally improves the functionality of both healthy and constipated gut microbiota by increasing SCFAs. Finally, the increase in AQP-3 expression suggests an additional mechanism of action for this plant extract. These results provide valuable insights into a non-pharmacological strategy for functional constipation.

## 2. Materials and Methods

### 2.1. H. rhamnoides Extract

Three batches of commercial, dry, powdered hydroethanolic extract of *H. rhamnoides* extract (standardized to contain a minimum of 0.10% isorhamnetin from the fruits, HPLC method, ref. Eur. Ph. 10.0, Monograph 1827), were provided by EPO S.R.L. (Milan, Italy). The extract was prepared by hydroethanolic extraction (50% ethanol), followed by concentration and evaporation solvent removal to dryness to obtain the standardized crude extract.

### 2.2. UHPLC Q-Orbitrap HRMS Analysis of H. rhamnoides Extract

The Vanquish UHPLC system (Thermo Fisher Scientific, Waltham, MA, USA) was used for the chromatographic analysis. The system comprised a thermostated column oven (TCC-3100, Thermo Fisher Scientific, Waltham, MA, USA), a refrigerated autosampler (WPS-3000, Thermo Fisher Scientific, Waltham, MA, USA), an in-line degasser (GPL-3400RS, Thermo Fisher Scientific, Waltham, MA, USA), and a binary solvent delivery pump (HPG-3400RS, Thermo Fisher Scientific, Waltham, MA, USA). At 30 °C, separation was accomplished on a Kinetex Biphenyl column (100 × 2.1 mm, 2.6 µm; Phenomenex, Torrance, CA, USA). Solvent A (0.1% formic acid in water) and solvent B (0.1% formic acid in methanol) were used as the mobile phase, which had a steady flow rate of 0.4 mL/min. The overall runtime was 13 min. The gradient program was the following: 100% A was maintained for 0.5 min, then linearly decreased to 30% A from 0.5 to 1.5 min, further reduced to 15% A from 1.5 to 8.0 min (corresponding to an increase in solvent B—0.1% formic acid in methanol—to enhance elution strength), and turned back to 100% A from 8.0 to 11.0 min, and maintained at that level for 2 additional min to wash and re-equilibrate the column. The injection volume was 5 µL. A Q-Exactive Orbitrap instrument (Thermo Fisher Scientific, Waltham, MA, USA), operated in negative mode, was used to perform high-resolution mass spectrometry. Data acquisition included full-scan MS followed by all-ion fragmentation (AIF). In full MS mode, data were collected over a mass-to-charge (*m*/*z*) range of 80–1200 with a resolving power of 70,000 full width at half maximum (FWHM). The acquisition rate was two scans/second, the maximum injection time was 0.2 s, and the automatic gain control (AGC) target was set at 1 × 10^6^. AIF acquisition was carried out with a resolution of 17,500 FWHM, an *m*/*z* range of 80–1200, and an AGC target of 1 × 10^5^. The scan duration was 0.1 s, and the maximum injection time was again set to 0.2 s. Fragmentation was induced using stepped normalized collision energies ranging between 15 and 45 eV. An isolation window of 5 *m*/*z* and a retention time integration window of 30 s were applied for AIF data collection. The ionization source parameters were as follows: spray voltage at 2.8 kV, capillary temperature of 275 °C, S-lens RF level set at 50, sheath gas (nitrogen, purity > 95%) at 35 arbitrary units, auxiliary gas at 10 units, and auxiliary gas heater temperature maintained at 350 °C. Compound identification followed the Metabolomics Standards Initiative (MSI) guidelines. Metabolites were annotated based on accurate mass (±5 ppm), MS/MS fragmentation pattern matching with spectral libraries, and comparison with authentic standards when available. Identifications were classified as Level 1 when confirmed by retention time, accurate mass, and MS^2^ spectra of standards; Level 2 for putatively annotated compounds based on accurate mass and MS^2^ spectral similarity; and Level 3 for putatively characterized compound classes showing characteristic fragmentation behavior but lacking full structural confirmation. MS^2^ data were used to support structural elucidation and annotation confidence. Mass accuracy was controlled with a tolerance of 5 ppm to ensure precise identification of analytes. Routine external calibration was performed before each analytical batch to ensure mass accuracy within ±5 ppm. Data processing and analysis were carried out using Xcalibur software (version 3.1.66.19, Thermo Fisher Scientific, Waltham, MA, USA) [17].

### 2.3. Evaluation of AQP-3 Expression in HT-29 Cells

#### 2.3.1. HT-29 (Intestinal Epithelial Cell Line) Culture Conditions

The human colorectal adenocarcinoma adherent epithelial cell line, HT-29, was obtained from the American Type Culture Collection (ATCC, Rockville, MD, USA). The cells were maintained in a humidified atmosphere with 5% CO_2_ at 37 °C, using Dulbecco’s Modified Eagle Medium (DMEM) supplemented with 10% fetal bovine serum and 1% penicillin–streptomycin.

#### 2.3.2. Cell Viability Determination

MTT (3-(4,5-dimethylthiazol-2-yl)-2,5-diphenyltetrazolium bromide) assay was employed to assess cell viability. A total of 2.5 × 10^4^ HT-29 cells per well were plated in a 96-well plate. The following day, cells were treated with increasing concentrations of *H. rhamnoides* fruit extract (2.1–210 ug/mL) and *H. rhamnoides*—maltodextrin combination (3–300 ug/mL). After 24 h, 25 µL of MTT solution (5 mg/mL in saline; Sigma-Aldrich, Milan, Italy) was added to each well, and the plate was incubated at 37 °C for 1.5 h. The resulting blue formazan crystals were dissolved in dimethyl sulfoxide (DMSO), and absorbance was measured at 490 nm using a microplate spectrophotometer (Multiskan FC, Thermo Scientific™, Waltham, MA, USA).

#### 2.3.3. Protein Extraction and Western Blot

Collected cells (5 × 10^5^/well) were lysed using RIPA (Thermo Fisher Scientific, Waltham, MA, USA) 1X lysis buffer 1X supplemented with 10 μL/mL phenylmethylsulfonyl fluoride (PMSF), 10 μL/mL sodium orthovanadate, and 10 μL/mL protease inhibitor and centrifuged at 14,000× *g* for 5 min at 4 °C. Protein concentration in the supernatant was determined by Bio-Rad protein assay (BioRad, Richmond, CA, USA), with bovine serum albumin (BSA) as the standard.

The 35 μg samples were loaded onto 8% SDS polyacrylamide gels and transferred to nitrocellulose membranes (Bio-Rad Laboratories Inc.). The membranes were then blocked with 5% non-fat dry milk for 1 h at room temperature and incubated overnight at 4 °C with primary antibody: anti-Aquaporin 3 (ab125219, Abcam (Cambridge, UK), 1:1000) and anti-Alpha Tubulin (11224-1-AP, Proteintech (Rosemont, IL, USA), 1:2000). Following primary antibody incubation, membranes were rinsed with PBS Tween 0.05% and incubated with secondary antibody, anti-rabbit (HAF008, R&D System, Minneapolis, MN, USA, 1:1000), for 2.5 h. After washing with PBS Tween, membranes were exposed to enhanced chemiluminescence (ECL) detection and imaged using the Chemidoc imaging system (Bio-Rad Laboratories Inc.). Image quantification was performed using Image Tool software 3.0.

### 2.4. In Vitro Digestion

*H. rhamnoides* fruit extract underwent in vitro gastrointestinal digestion to simulate physiological human intestinal processes, according to Minekus et al. [18], with minor changes [19]. Briefly, 5 g of extract was mixed with 5 mL of simulated salivary fluid (SSF), followed by the addition of 0.5 mL of fresh α-amylase solution (150 U/mL). Following incubation for 2 min at 37 °C in a shaking water bath, 10 mL of simulated gastric fluid (SGF) containing pepsin (4000 U/mL; pH 3.00) was added. The final volume was adjusted to 20 mL, and the mixture was allowed to incubate for 2 h at 37 °C in a shaking water bath. Finally, gastric chyme was mixed with 5 mL of freshly prepared pancreatin (800 U/mL) and 2.5 mL. Gastric chyme was mixed with 20 mL of simulated intestinal fluid containing freshly prepared pancreatin (200 U/mL) and bile salts (20 mM). The final volume was adjusted to 40 mL (pH 7.00), and the mixture was incubated for 2 hrs at 37 °C in a shaking water bath. Enzyme activity was stopped at the end of each step by immersion of tubes in ice for 15 min. The supernatant at each digestion step was collected and stored at −20 °C for subsequent analysis of the antioxidant capacity, while the pellet (i.e., the undigested fraction) was directly used for in vitro fermentation. Three independent in vitro digestions were performed on separate aliquots of the extract (biological replicates, *n* = 3). Antioxidant assays on digested supernatants were measured in technical triplicate for each biological replicate.

### 2.5. In Vitro Fermentation

In vitro fermentation of the digested *H. rhamnoides* fruit extract was carried out according to Pérez-Burillo et al. [20]. Adult donors were healthy Caucasian and constipated subjects. Since samples collected from human subjects were used, the study adhered to the guidelines of the Declaration of Helsinki and was approved by the Ethics Committee of the University of Granada (protocol code 1080/CEIH/2020) for the adult participants. None of the donors had taken antibiotics within the previous three months. For each arm (healthy and constipated), fecal material from three donors was pooled to prepare a single standardized inoculum (the donor is not the unit of replication). Using this pooled inoculum, we carried out three independent batch fermentations per arm (biological replicates, *n* = 3). Centrifuge tubes containing 10% of the final digestion volume and 500 mg of the solid digestion residue were supplemented with 7.5 mL fermentation growth medium (peptone (14 g/L), cysteine (312 mg/L), sodium sulfide (312 mg/L), and resazurin (0.1% *v*/*v*)) and 2 mL fecal inoculum, which was prepared by suspending the fecal material in phosphate-buffered saline, 33% *w*/*v*. Nitrogen was bubbled into the tubes to maintain anerobic conditions. The tubes were incubated under oscillation for 20 h at 37 °C, and finally placed on ice for 15 min to stop microbial activity. Sample centrifugation (6000× *g* for 10 min) allowed separation and collection of supernatant, which was stored at −20 °C for analysis of antioxidant capacity and SCFAs (technical triplicates per biological replicate) and SCFA analysis.

### 2.6. Antioxidant Capacity Profiling

#### 2.6.1. TPC Determination

The Folin–Ciocalteu assay for the determination of the TPC was carried out according to Moreno-Montoro et al. [21]. Briefly, 30 µL of the digestion (or fermentation supernatant) was mixed with 15 µL of Folin–Ciocalteu reagent, 60 µL of sodium carbonate (10% *w*/*v*), and 195 µL of ultra-pure water in a 96-well plate (Cytation, Agilent Technologies Inc., Santa Clara, CA, USA). The mixture was incubated for 60 min at 37 °C. Gallic acid (0.01 to 1.00 mg/mL) was used as a standard to draw a calibration curve. The results were expressed as mg gallic acid/100 g (mg GAE/100 g) of the *H. rhamnoides* extract. Although the Folin–Ciocalteu assay is not fully specific for phenolic compounds and may also respond to other reducing agents present in digested/fermented matrices, it was selected here because it is the most widely adopted colorimetric method in this research area and therefore allows direct comparability of our data with previously published studies.

#### 2.6.2. 2,2-Diphenyl-1-Picrylhydrazyl (DPPH) Assay

The DPPH assay of the digested and fermented samples was performed according to Yen and Chen [22]. Supernatants of digested or fermented samples were mixed with the 280 µL DPPH reagent, prepared freshly by dissolving 74 mg DPPH in 1 L methanol. The antioxidant reaction was monitored for 1 h at 37 °C using a Cytation 5 plate reader (Agilent Technologies Inc., Santa Clara, CA, USA). Trolox (0.01–0.4 mg/mL) was used as a standard for the calibration curve. Results were expressed as mmol Trolox equivalents/kg (mmol TE/kg) of *H. rhamnoides* extract.

#### 2.6.3. 2,2′-Azino-Bis(3-Ethylbenzothiazoline-6-Sulfonic Acid) (ABTS) Assay

The ABTS^+^ assay of the digested and fermented samples was carried out according to Re et al. [23] with slight modifications. A 7 mM ABTS^+^ stock solution was mixed with 2.45 mM potassium persulfate to prepare the ABTS^+^ solution, which was allowed to stand in the dark at room temperature for 16 h. Before use, the ABTS^+^ solution was diluted with 5 mM phosphate-buffered saline (pH 7.4) before the assay to achieve an absorbance of 0.70 + 0.02 at 730 nm. For the assay, 10 µL of the supernatant from digested or fermented samples was added to 4 mL of diluted ABTS^+^ solution and incubated for 20 min. The reading absorbance was at 730 nm. The calibration curve was prepared using the Trolox stock solution, and the results were expressed as mmol TE/kg of *H. rhamnoides* extract.

#### 2.6.4. FRAP Assay

The FRAP assay of the digested and fermented samples was performed using protocols set by Benzie and Strain [24], with slight modifications. Samples (20 µL) were mixed with freshly prepared FRAP reagent (280 µL) in a 96-well plate. The FRAP reagent was prepared by mixing 40 mM 2,4,6-tri(2-pyridyl)-s-triazine (TPTZ) solution, ferric chloride solution, and 0.3 M sodium acetate buffer (pH 3.6) in a ratio of 1:1:10. The assay was performed at 37 °C and the absorbance was measured at 595 nm every 30 s for 30 min. Trolox (0.01–0.4 mg/mL) was used as a standard to prepare a calibration curve. Results were expressed as mmol TE/kg of *H. rhamnoides* extract.

### 2.7. SCFA Analysis

SCFAs were analyzed in supernatants from fermentation samples according to Panzella et al. [25]. For chromatographic analysis, the samples were centrifuged at 13,300 rpm for 5 min, filtered through a 0.22 µm filter and diluted with 1 M HCl acid in a 1:10 ratio. The analysis was performed under isocratic elution (at 0.5 mL/min flow rate) using an Agilent Poroshell 120 SB-Aq column (3 × 150 mm, 2.7 µm). Sulfuric acid (5 mM) was used as a mobile phase, and a 5 µL sample volume was injected. The temperature of both the column and the refractive index detector was maintained at 35 °C. External standards (propionic, n-butyric, isobutyric acids, lactic acid, and succinic acid) were used to prepare a calibration curve to quantify SCFAs. n-butyrate and isobutyrate were quantified together, while total SCFA was calculated as the sum of the acetate, propionate, and butyrate. SCFAs were quantified from the three fermentation supernatants per arm (biological *n* = 3). Each sample was injected in duplicate, and duplicates were averaged prior to statistical analysis. Results were expressed as mmol of each acid/liter of the fermented sample.

### 2.8. Statistical Analysis

Statistical comparison between the groups was carried out as follows. First, the antioxidant capacity (TPC, TEAC_ABTS_, TEAC_FRAP_, and TEAC_DPPH_) of the *H. rhamnoides* fruit extract was evaluated by comparing non-digested samples with those subjected to oro-gastroduodenal digestion. Samples fermented with fecal microbiota (isolated from constipated adult subjects) were compared with those from healthy adults to assess changes in TPC values and antioxidant capacity under healthy and poor conditions. These comparisons were also carried out for samples exposed to duodenal digestion. The increase in SCFAs was assessed by comparing each metabolite produced during fermentation of *H. rhamnoides* fruit extract by healthy versus constipated microbiota. For these analyses, unpaired *t*-tests were applied; when the assumption of homogeneity of variance was not met, Welch’s correction was used. To minimize the risk of type I errors due to multiple testing, Bonferroni’s correction was applied.

The significance of the differences between treated samples and untreated cells in the cell viability assay and Western blot analysis was determined using one-way ANOVA followed by Dunnett’s test, with GraphPad Software Prism v9.0 (San Diego, CA, USA).

## 3. Results

### 3.1. Metabolite Profile of H. rhamnoides Fruit Extract

The first step was the evaluation of the metabolite profiling of the commercial hydroethanolic *H. rhamnoides* fruit extract. Both positive and negative electrospray ionization modes during method development were evaluated. However, the compounds of interest in *H. rhamnoides* extract—primarily phenolic acids and flavonoid—exhibited significantly better ionization efficiency and signal intensity in the negative ion mode, predominantly forming [M–H]^−^ ions. In contrast, ionization in the positive mode resulted in weaker signals and lower sensitivity for most detected metabolites. Hence, we selected negative mode for the final LC–MS analysis to ensure optimal detection and coverage of the characteristic metabolite profile. The isolation window of 5 *m*/*z* was selected as a compromise between selectivity and sensitivity, in line with the resolution settings of the instrument and the complexity of the extract matrix. The main classes are flavonoids (Table 1 flavonols, particularly isorhamnetin and its glycosides, as well as quercetin derivatives; flavones, apigenin and glycosides; a flavanol, epicatechin; and a flavanone, naringenin) and phenolic acids (hydroxybenzoic and hydroxycinnamic acids: protocatechuic, syringic, caffeic, and p-coumaric), together with ellagic acid. In terms of abundance (peak area), quinic acid is the most abundant overall; among polyphenols, protocatechuic acid and isorhamnetin glycosides predominate.

**Table 1 foods-14-03800-t001:** Tentatively identified compounds in the hydroethanolic fruit extract by UHPLC Q-Orbitrap HRMS (negative ion mode) and their corresponding analytical parameters (*n* = 23). * columns report no.; compound name; RT (min); molecular formula; adduct/ion; theoretical *m*/*z*; measured *m*/*z*; mass accuracy (Δ, ppm); MSI level (1 = confirmed with authentic standard, 2 = putatively annotated by MS/MS spectral match).

No.	Compound	Retention Time (RT)	Molecular Formula	Adduct/ Ion	Theoretical Mass (*m*/*z*)	Measured Mass (*m*/*z*)	Accuracy (Δ)	MSI Level
1	Apigenin	5.28	C_15_H_10_O_5_	[M-H]^−^	269.04555	269.0457	0.3717	1
2	Apigenin 7-glucoside	5.03	C_21_H_20_O_10_	[M-H]^−^	431.09837	431.0985	0.2552	1
3	Caffeic acid	4.61	C_9_H_8_O_4_	[M-H]^−^	179.03403	179.0342	1.0054	1
4	Ellagic acid	4.57	C_14_H_6_O_8_	[M-H]^−^	300.99899	300.9991	0.3987	1
5	Epicatechin	4.16	C_15_H_14_O_6_	[M-H]^−^	289.07176	289.0723	1.7643	1
6	Isorhamnetin-3-rutinoside	4.59	C_28_O_32_O_16_	[M-H]^−^	623.16176	623.1622	0.6579	1
7	Kaempferol-3-O-glucoside	4.61	C_21_H_20_O_11_	[M-H]^−^	447.09328	447.0936	0.7157	1
8	Kaempferol	5.12	C_15_H_10_O_6_	[M-H]^−^	285.04046	285.0404	−0.0702	1
9	Luteolin-7-glucoside	4.78	C_21_H_20_O_11_	[M-H]^−^	447.09328	447.0938	1.1183	1
10	Naringenin	5.22	C_15_H_12_O_5_	[M-H]^−^	271.06119	271.0612	1.3650	1
11	*p*-coumaric acid	4.36	C_9_H_8_O_3_	[M-H]^−^	163.04006	163.0391	−2.7601	1
12	Protocatechuic acid	2.83	C_7_H_6_O_4_	[M-H]^−^	153.01933	153.0182	−0.7189	1
13	Quercetin	4.85	C_15_H_10_O_7_	[M-H]^−^	301.03537	301.0359	1.7274	1
14	Quercetin 3b glucoside	4.46	C_21_H_20_O_12_	[M-H]^−^	463.08819	463.0889	1.4684	1
15	Quinic acid	0.65	C_7_H_12_O_6_	[M-H]^−^	191.05528	191.0562	4.9200	1
16	Rutin	4.40	C_27_H_30_O_16_	[M-H]^−^	609.14610	609.1461	0.0657	1
17	Syringic acid	4.30	C_9_H_10_O_5_	[M-H]^−^	197.04510	197.04490	−1.2180	1
18	Isorhamnetin	5.29	C_16_H_12_O_7_	[M-H]^−^	315.05103	315.05140	1.0474	2
19	Isorhamnetin 3-rhamnoside	5.20	C_22_H_22_O_11_	[M-H]^−^	461.10893	461.10920	0.6289	2
20	Isorhamnetin O-hexoside	4.65	C_22_H_22_O_12_	[M-H]^−^	477.10384	477.10420	0.7336	2
21	Apigenin-O-hexosy-l-6-C-hexoside	4.55	C_27_H_30_O_15_	[M-H]^−^	593.15119	593.15230	1.9388	2
22	Quercetin 3-glucoside-7-rhamnoside	4.26	C_27_H_30_O_16_	[M-H]^−^	609.14610	609.14700	1.4282	2
23	Isorhamnetin-3-rutinoside isomer 2	4.39	C_28_H_32_O_16_	[M-H]^−^	623.16176	623.16230	0.8505	2

MS/MS fragmentation data (MS^2^) were used for the structural elucidation and to support the annotations, particularly for Level 1 and Level 2 identifications. Where available, authentic standards were employed to validate compound identity.

### 3.2. Bioaccessibility of H. rhamnoides Fruit Extract After In Vitro Simulated Oro-Gastroduodenal Digestion and Fermentation Processes

#### 3.2.1. Impact on *H. rhamnoides* Polyphenol Stability

The extract was subjected to in vitro simulated oro-gastroduodenal digestion and fermentation using the gut microbiota isolated from healthy and constipated adults to test the effect of digestion and microbial fermentation on individual and total polyphenols. The extract was then analyzed using UHPLC Q-Orbitrap HRMS before and after digestion and fermentation. Table 2 shows the total and % areas of the chromatographic peaks of the identified compounds occurring in the extract after oro-gastroduodenal digestion before (OGDd) and after digestion and fermentation with gut microbiota isolated from healthy (OGDd + healthy fermentation) and constipated (OGDd + constipated fermentation) adults. All in all, oro-gastroduodenal digestion induced a 10% increase in polyphenol total peak areas, the highest % increase being found in apigenin aglycone, maybe due to the hydrolysis of glycosidic bonds of apigenin glycoside derivatives, and *p*-coumaric acid, likely released as a free polyphenol by the food matrix during digestion. Microbial fermentation also showed a 10% and 70% loss of polyphenol total peak areas in healthy and constipated groups, respectively, suggesting a different metabolic end for the polyphenols when interacting with the gut microbiota isolated from the fecal material of constipated and healthy subjects. These results were confirmed by the TPC measurement (Table 3). The decrease in TPC of the extract (982.6 ± 19.9 mg GAE/100 g after digestion and fermentation using gut microbiota from healthy subjects and 587.3 ± 46.6 mg GAE/100 g, *p* < 0.01, after digestion and fermentation with gut microbiota from constipated subjects) argues for higher than normal microbial metabolism of polyphenols in constipation.

#### 3.2.2. Antioxidant Properties of *H. rhamnoides* Fruit Extract, Before and After Digestion and Fermentation

Determination of the residual antioxidant properties of the extract provides an estimate of the radical scavenging and reducing capacity of the extract during the digestion and fermentation processes, with the gastrointestinal tract being a major site of reactive oxygen species (ROS) formation. We selected the DPPH, ABTS, and FRAP assays to comprehensively evaluate the antioxidant capacity of the *H. rhamnoides* extract, as these methods provide complementary information regarding different antioxidant mechanisms. DPPH and ABTS assays both assess radical scavenging capacity (DPPH primarily captures activity of lipophilic antioxidants, while ABTS also encompasses hydrophilic compounds). In contrast, FRAP measures the ability of antioxidants to reduce ferric (Fe^3+^) to ferrous (Fe^2+^) ions, reflecting the reductive potential of the sample [26]. The combined use of DPPH, ABTS, and FRAP provides methodological advantages, as it offsets the limitations associated with any single assay and ensures sensitivity to a wider range of antioxidant molecules present in complex botanical extracts. DPPH is sensitive to smaller, nonpolar antioxidants; ABTS detects both hydrophilic and lipophilic compounds; and FRAP assesses the electron-donating capacity relevant to physiological oxidation-reduction reactions. Statistically significant differences between treatment conditions were obtained (Table 3).

In terms of the radical scavenging activity, digestion processes did not modify the hydrogen donating ability of the *H. rhamnoides* extract, as shown by statistically unchanged DPPH values before and after digestion (201.8 ± 5.3 vs. 202.5 ± 6.6 mmol TE/kg; Table 3). Following fermentation, DPPH values decreased both in healthy (30.6 ± 1.7 mmol TE/kg) and constipated (45.9 ± 4.9 mmol TE/kg) groups, with differences across the groups being not statistically significant. After oro-gastroduodenal digestion, ABTS^+^ values were increased significantly (from 35.5 ± 2.7 to 59.3 ± 0.7 mmol TE/kg, *p* < 0.01; Table 3). This suggests the release of bound antioxidant compounds from the food matrix (or the conversion of polyphenols into more active forms during digestion processes). ABTS^+^ activity was decreased post-fermentation (21.2 ± 1.1 mmol TE/kg and 15.9 ± 0.6 mmol TE/kg in healthy and constipated groups, respectively). However, consistent with diversity in gut microbiota and microbial metabolism of constipated subjects, the healthy microbiota group had higher residual ABTS^+^ activity, while the constipated microbiota showed greater use of radical scavengers. On the other hand, consistent with a partial loss of reducing power of *H. rhamnoides* fruit extract following digestion, FRAP values were decreased to 145.4 ± 3.8 mmol TE/kg after digestion (values before digestion being 194.6 ± 30.8 mmol TE/kg, *p* < 0.05; Table 3). Following fermentation, FRAP activity was decreased to 2.94 ± 0 mmol TE/kg in healthy adults and was not detected in constipated adults (*p* < 0.0001), indicating near-to-complete loss of ferric reducing potential.

### 3.3. SCFA Analysis

SCFAs are the primary metabolites of intestinal microbiota. They are important for the homeostasis of the intestinal environment and promote normal intestinal regularity, being able to stimulate water and electrolyte absorption and to affect gastrointestinal motility. SCFA concentration was determined by UHPLC-RID analysis to evaluate the impact of digested *H. rhamnoides* fruit extract on the functionality of gut microbiota isolated from healthy and constipated donors. The results (Figure 1) showed a higher concentration of SCFAs produced in response to *H. rhamnoides* fruit extract in the healthy group (56.02 mM) than in the constipated group (48.57 mM). In particular, lactate, propionate, and butyrate were produced in higher concentrations by the healthy microbiota, whereas acetate showed higher production by the microbiota isolated from constipated donors.

### 3.4. Intestinal Stimulatory Effect of H. rhamnoides Fruit Extract on Aquaporin-3 Expression

Aquaporins (AQPs) play an important role in the water transport system of the human body. AQP-3 is predominantly expressed in the colon, ultimately controlling water transport. Several products exhibit a laxative effect by changing the expression level of AQP-3 in the colon. Based on these reports, AQP-3 is hypothesized to play a key role in water transport in the colon. We evaluated the effect of the treatment of HT-29 cells with non-cytotoxic concentrations of *H. rhamnoides* fruit extract on AQP-3 expression.

#### 3.4.1. Effect of *H. rhamnoides* Fruit Extract on Cell Viability

As shown in Figure 2, at concentrations ranging from 2.1 to 210 µg/mL, the treatment with *H. rhamnoides* fruit extract had no significant effects on the viability of HT-29 cells when compared to the vehicle control (10% methanol/water). Hence, *H. rhamnoides* fruit extract is non-cytotoxic to HT-29 cells in these conditions.

#### 3.4.2. Effect of *H. rhamnoides* Fruit Extract on AQP-3 Expression

As shown in Figure 3 and Appendix A, treatment of HT-29 cells with *H. rhamnoides* fruit extract at concentrations ranging from 7 to 70 µg/mL for 24 h resulted in a significant upregulation of AQP-3 expression compared to the vehicle control (10% methanol/water). These results suggest that *H. rhamnoides* extracts enhance AQP-3 expression in HT-29 cells, indicating their potential role in modulating AQP-3-related cellular functions.

## 4. Discussion

Oxidative stress is an important risk factor for colonic dysmotility, which can lead to constipation [27]. Polyphenols exert antioxidant and radical scavenging activities. Since the small intestine is known to absorb polyphenols poorly [28], a large quantity of dietary polyphenols reaches the colon, where they are metabolized by the colonic microbiota [29], with the potential to confer health benefits via modulation of the gut micro-ecology [30]. Its disruption (dysbiosis) can lead to functional gastrointestinal disorders and, in turn, constipation [31]. Thus, prior to assessing bioaccessibility and the biological effects of *H. rhamnoides* fruit extract, this study was aimed at evaluating its polyphenol profiling using UHPLC Q-Orbitrap HRMS analysis. A rich profile of polyphenols, including both aglycones and glycosides, was revealed by the present study. Among the phenolic acids, protocatechuic acid and caffeic acid exhibit higher peak areas, indicating their predominance. Notably, caffeic acid maintains considerable levels after digestion and fermentation, reflecting its stability and potential bioactivity. Within the flavonol class, the extract revealed high concentrations of isorhamnetin and its derivatives (isorhamnetin-3-rutinoside, isorhamnetin 3-rhamnoside, and isorhamnetin O-hexoside). These products persist relatively abundantly following digestion and fermentation. Quercetin and its glucoside are also present in substantial amounts, although their levels somehow decrease post-fermentation, suggesting moderate stability. Other significant flavonoids include apigenin 7-glucoside and luteolin-7-glucoside, further confirming the richness of flavones in the extract. Our findings align with a comprehensive review that documented the presence of polyphenols from different chemical classes in *H. rhamnoides* [12]. Another study in which UPLC-PDA-Q/TOF-MS was employed confirmed the presence of phenolic acids, catechins, and flavonol derivatives in an *H. rhamnoides* fruit extracts. [32].

Although being beneficial for GI health, ROS are potentially involved in the pathogenesis of intestinal disorders, including constipation, due to the oxidative stress generated by an imbalance between their production and catabolism. Thus, we evaluated the antioxidant properties of *H. rhamnoides* fruit extract before and after the in vitro simulated digestion process and fermentation by fecal microbiota through the determination of the total polyphenol content, DPPH and ABTS radical scavenging activity and reducing capacity. Our results are at variance with most available literature. Li et al. reported a slightly higher (from 32.20 to 33.51 mg GAE/g TPC) fruit extract for *H. rhamnoides* [32]. Interestingly, our TPC values are higher than those by Ursache et al. with respect to nine different *H. rhamnoides* cultivars from Slovakia, where TPC ranged from 5.19 to 23.97 mg GAE/g [33]. Other studies have reported significantly higher TPC in different parts of the *H. rhamnoides* plant, with values ranging from 84.94 to 302.72 mg GAE/g [34,35]. The DPPH, ABTS, and FRAP values in our study were also lower than those reported in the literature. He et al. [36] tested different *H. rhamnoides* leaf tea extracts and reported DPPH, ABTS, and FRAP values of 20.69–50.21, 0.81–9.84, and 12–17.94 mmol Trolox equivalent/g, respectively. This suggests that leaves may possess a more potent antioxidant profile compared to fruits in this setting. The possibility is now under investigation that differences are likely due to differing extraction procedures, geographical location, or genetic backgrounds of the plants.

In vitro digestion of the *H. rhamnoides* fruit extract led to a non-significant decrease in its TPC (from 2924.2 mg GAE/100 g to 2235.7 mg GAE/100 g) and a slight increase in the total chromatographic peak area of the selected polyphenols. The DPPH radical scavenging activity persisted stable after digestion (201.8 mmol Trolox equivalent/kg before vs. 202.5 mmol Trolox equivalent/kg after digestion). ABTS values increased significantly after digestion (*p* < 0.05), vis-à-vis the significant decrease (*p* < 0.05) of the reducing potential. The data on the content of polyphenols show that some compounds are unstable in the gastric and intestinal environments. Pancreatic enzymes and bile salts can hydrolyze glycosidic bonds, releasing aglycones, which are more susceptible to degradation by digestive enzymes and pH changes [37,38,39,40]. However, digestion can also release polyphenols from their food matrix, thus partially offsetting the loss [41]. The release of these bound polyphenols and the hydrolysis of glycosides into aglycones likely explain why DPPH and ABTS values were not as affected as the TPC. Since aglycones are potent hydrogen donors, the DPPH values were unchanged [42]. Where DPPH is more responsive to lipophilic compounds, ABTS can react with both hydrophilic and lipophilic phenols [23,43].

Fermentation of the digested *H. rhamnoides* fruit extract was performed using fecal samples from three donors. The pooling of fecal samples from three donors per group is a common practice to average inter-individual variability in in vitro fermentation studies. However, the main limitation of this choice, i.e., the loss of information on individual variation in microbial response, should be acknowledged. The fermentation resulted in a significant loss of total polyphenols, as measured both spectrophotometrically and chromatographically, as well as antioxidant capacity. This effect was more pronounced when fermentation was induced by the gut microbiota isolated from the fecal materials of constipated subjects. Evidence suggests that the gut microbiota degrades complex polyphenols into simpler undetectable metabolites [44,45]. This polyphenol loss directly reduced the antioxidant capacity, as measured by DPPH, ABTS, and reducing power. Additionally, gut microbiota consumes hydrogen donors as metabolic substrates during fermentation and, in turn, decreases DPPH activity [46,47]. While a decreased DPPH activity was found using samples from both healthy and constipated subjects, the effect was more pronounced in the samples from the healthy individuals. This is likely related to a higher metabolic conversion due to a diverse, healthy intestinal microbiota in these settings. The ABTS assay argued for a slightly greater preserving effect in fecal samples from healthy subjects, suggesting that the microbiota in constipated individuals transform radical scavengers into less active forms and/or deplete them more rapidly. The complete loss of FRAP activity is likely due to the extensive degradation of electron-rich polyphenols, e.g., quercetin and catechins, during fermentation, with smaller compounds with negligible reducing power persisting [48,49].

In addition to the direct radical scavenging activity of the *H. rhamnoides* fruit extract, the effect of the extract following digestion and fermentation with fecal bacteria isolated from healthy and constipated adults on SCFA production was investigated to unravel its impact on gut microbiota functionality. Some investigations have documented that the microbiota exerts a crucial influence on gut motility mediated primarily by SCFA metabolites [50,51,52]. We show that the levels of acetate, butyrate, lactate, succinate, and propionate variably increase in the presence of gut microbiota isolated from both healthy and constipated subjects after exposure to *H. rhamnoides* fruit extract. All in all, SCFA concentrations in samples from healthy individuals were higher (56.0 mM) than those from constipated subjects (48.6 mM). These results are interesting, as SCFA concentrations are low in subjects with constipation [53]. Some studies suggest that propionic and butyric acid affect intestinal motility by directly acting on colonic smooth muscles [54], and that SCFA can reduce the intestinal transit time by increasing the concentration of serotonin (5-HT) in the gut [55]. Specifically, by stimulating 5-HT release [56], butyrate reduces constipation by increasing intestinal motility. In keeping with this, 5-HT levels in constipated subjects are often lower than normal subjects, causing slowed transit and hardened stools [57,58]. SCFA concentration increases suggest improved functionality of the gut microbiota as well as of its composition in support of microbial species producing SCFA. In this study, a potential reduction in the diversity of lactate- and butyrate-producing bacteria may have shifted fermentation pathways toward acetate as the predominant SCFA present when fermentation is induced by fecal microbiota isolated from constipated adults. Growing evidence suggests that an abnormal gut microbe (dysbiosis) contributes to functional constipation by modifying colonic motility, secretion, and absorption via microbial metabolites such as methane, 5-hydroxytryptamine, bile acids, and SCFA [59]. Dysbiosis in constipation is often characterized by a high concentration of methanogens and a low presence of lactate- and butyrate-producing bacteria [60]. Zoppi et al. [61] found that constipated children had higher levels of Bifidobacteria and Clostridium compared to healthy controls. Khalif et al. observed a similar trend in adults [62]. Using quantitative real-time PCR, Kim et al. showed that feces from patients with functional constipation had abnormally low species of Bacteroides and Bifidobacterium [63]. Using whole-genome and 16S rRNA sequencing, the gut microbial composition of adults with functional constipation was consistent with lower SCFA synthesis [64]. Acetate is physiologically consumed by butyrate-producing bacteria. The lack of these cross-feeding bacteria in constipation might accumulate acetate [65].

Recent studies have proposed AQPs as a therapeutic target in both diarrhea and constipation [66]. AQPs are specialized membrane proteins acting as channels to facilitate water and small, neutral solute transport across cell membranes. Of the 13 known types (AQP0-AQP12), AQP-3 is highly expressed in the colon and plays a key role in water transport, making it a key target for handling constipation [66]. Modulating AQP-3 expression could relieve constipation by enhancing stool hydration. We find that the *H. rhamnoides* fruit extract (at 7–70 µg/mL concentrations) upregulates AQP-3 expression in HT-29 cells. The regulation of AQP-3 expression by plant and food extracts is predicted to relieve constipation. Soluble fibers from hawthorn (*Crataegus monogyna* Jacq.) alleviate loperamide-induced constipation in mice by improving gut microbiome and modulating AQP expression. [67]. Likewise, naringenin alleviates loperamide-induced constipation in mice by upregulating AQP-3 expression [68]. On the other hand, consistent with the possibility that pathways other than AQP-3 modulation are involved in constipation relief, a fermented rice extract alleviates loperamide-induced constipation while reducing AQP-3 expression and oxidative stress and suppressing inflammation by altering MAPK phosphorylation [69].

## 5. Conclusions

The *H. rhamnoides* fruit extract examined in this study emerged as a rich source of bioactive compounds able to modulate the gut microbiota and AQP-3 membrane protein channel expression. Following in vitro digestion, our study found that the polyphenol content and DPPH radical scavenging activity remained unchanged, the reducing capacity of the extract decreased, and its ABTS values increased. This is tentatively due to the different sensitivity of different antioxidant compounds to different assays and the balance between the degradation of polyphenols, caused by the gastric and intestinal environment and the action of digestive enzymes and bile salts (offset by the release of some polyphenols from the vegetable matrix). When the digested extract was fermented with fecal samples, a key finding was the variability in total polyphenols and antioxidant capacity between samples from healthy and constipated subjects. This difference is accounted for by the diversity in gut microbiota and microbial metabolism of the constipated group. The modulation of SCFA synthesis and AQP-3 expression by the *H. rhamnoides* fruit extract suggests a plausible direction to be pursued to understand the mechanisms underlying the traditional use of sea buckthorn as a constipation remedy.

This study has several limitations. The simplified in vitro digestion model, while useful, cannot fully replicate the complex digestive and absorptive processes in humans. The practice of pooling fecal samples masks individual differences in microbial response. The biological effects observed lack in vivo validation. Accordingly, these findings remain preliminary, and human studies are mandatory to account for factors like intestinal motility, osmotic gradients, and hormonal effects. To address these issues and to validate our preliminary preclinical results and provide a more comprehensive understanding of the extract’s effects, a randomized, double-blind, placebo-controlled, parallel-arm clinical trial is in progress in subjects with functional constipation diagnosed with Rome IV criteria (https://clinicaltrials.gov/search?spons=epo%20, accessed on 15 October 2025).

## Figures and Tables

**Figure 1 foods-14-03800-f001:**
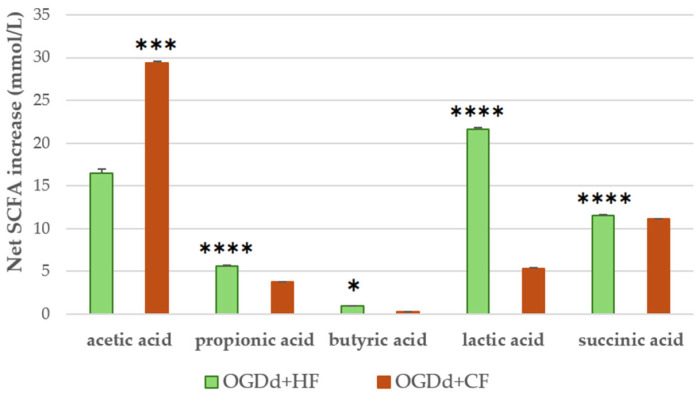
SCFAs produced by gut microbiota from healthy and constipated adults after fermentation of the digested *H. rhamnoides* fruit extract. The y-axis shows the net increase in SCFAs, calculated as the sample minus its blank (fecal fermentation without extract). The values are expressed in mmol/L, determined by subtracting the SCFA values of the blank controls (fecal samples fermented without the *H. rhamnoides* extract) from those of the *H. rhamnoides*-containing samples. Data are presented as the means ± SD. Statistics between group comparisons were performed by unpaired *t*-tests (Welch’s correction when needed) with Bonferroni adjustment for multiple testing; * *p* < 0.05, *** *p* < 0.001, and **** *p* < 0.0001.

**Figure 2 foods-14-03800-f002:**
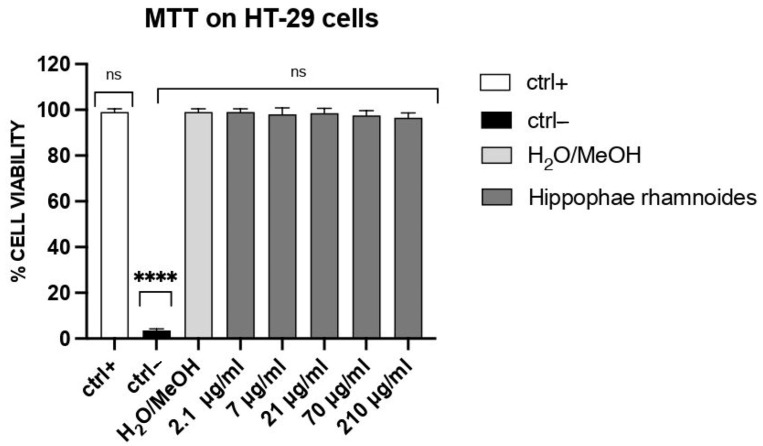
Effect of *H. rhamnoides* fruit extract on HT-29 cell viability. Cells were treated for 24 h with the hydroethanolic extract (2.1–210 μg/mL) or vehicle (10/90 methanol/water). Cell viability was evaluated by the mitochondrial-dependent reduction in MTT to formazan. Cell viability was assessed by the mitochondrial-dependent reduction in MTT to formazan. The y-axis reports show % cell viability (mean ± SD, *n* = 3 independent experiments) calculated from MTT absorbance at 490 nm and normalized to Ctrl+ (set to 100%). Groups: Ctrl+ = untreated cells (no solvent, no extract); Ctrl− = 100% DMSO (cytotoxic control). Statistics: one-way ANOVA followed by Dunnett’s test vs. ctrl+ (biological replicates *n* = 3; technical triplicates averaged); **** *p* < 0.0001; ns: not significant.

**Figure 3 foods-14-03800-f003:**
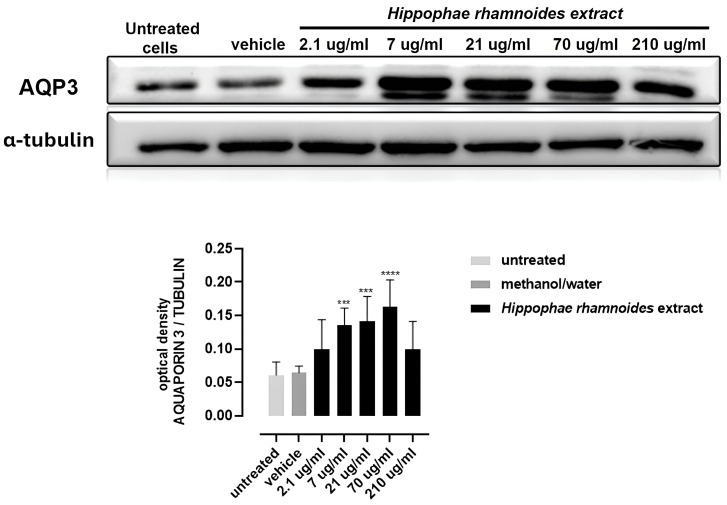
AQP-3 expression in HT-29 cells after 24 h treatment with *H. rhamnoides* fruit extract (2.1–210 μg/mL) or vehicle (10/90 methanol/water). Representative Western blots are shown; quantification is reported as the AQP-3/α-tubulin ratio. Statistics: one-way ANOVA with Dunnett’s post-hoc vs. vehicle; *** *p* < 0.001, **** *p* < 0.0001.

**Table 2 foods-14-03800-t002:** Impact of simulated oro-gastroduodenal digestion and microbial fermentation with gut microbiota isolated from the fecal materials of healthy and constipated adults on the composition of *H. rhamnoides* extract determined by UHPLC–HRMS. Extract = undigested extract; OGDd = after oro-gastroduodenal digestion; OGDd + healthy fermentation and OGDd + constipated fermentation = after subsequent batch fermentation with fecal microbiota from healthy or constipated adults. Peak area is reported in arbitrary units. % columns indicate the percentage of each compound’s peak area relative to the corresponding condition’s total peak area (* % after digestion; ** % after digestion + fermentation). Totals at the bottom give the sum of peak areas and % of total area vs. the undigested extract.

Compound	Extract	OGDd		OGDd + Healthy Fermentation		OGDd + Constipated Fermentation	
	Peak Area	Peak Area	% *	Peak Area	% **	Peak Area	% **
Apigenin	76 4,449	3,585,579	469	251,837	33	245,638	32
Apigenin 7-glucoside	58,124,033	74,123,957	128	-	-	-	-
Caffeic acid	186,219,833	135,329,204	73	24,684,355	13	20,598,877	11
Ellagic acid	53,430,815	7,121,377	13	2,283,194	4	1,758,333	3
Epicatechin	10,388,184	7,738,589	74	231,339	2	-	-
Isorhamnetin-3-rutinoside	526,701,724	561,802,160	107	440,88	-	525,187	
Kaempferol-3-O-glucoside	6,113,790	13,573,552	222	-	-	-	-
Kaempferol	32,240,801	26,524,808	82	6,080,500	19	1,2676,549	39
Luteolin-7-glucoside	44,000,357	47,449,461	108	-	-	-	-
Naringenin	9,635,719	3,144,428	33	-	-	-	-
*p*-coumaric acid	15,667,015	58,505,710	373	6,090,097	39	2,729,409	17
Protocatechuic acid	653,822,251	587,967,397	90	142,184,305	22	153,486,892	23
Quercetin	89,848,197	66,872,166	74	2,437,755	3	9,147,528	10
Quercetin 3b glucoside	48,934,260	38,834,407	79	-	-	-	-
Quinic acid	11,992,365,593	13,776,026,003	115	14,122,718,903	118	4,230,495,179	35
Rutin	41,976,587	26,302,844	63	-	-	-	-
Syrngic acid	3,979,140	4,354,804	109	-	-	-	-
Isorhamnetin	1,015,697,080	819,638,512	81	121,932,130	12	322,293,781	32
Isorhamnetin-3-rhamnoside	338,481,439	392,946,764	116	2,398,967	1	1,226,830	-
Isorhamnetin-O-hexoside	264,713,668	309,070,065	117	46,390	-	832,470	-
Apigenin-O-hexosy-l-6-C-hexoside	17,554,150	15,805,754	90	-	-	-	-
Quercetin 3-glucoside-7-rhamnoside	3,836,755	5,314,855	139	-	-	-	-
Isorhamnetin-3-rutinoside isomer 2	87,761,236	120,142,322	137	-	-	-	-
total area	15,502,257,076	17,102,174,718		14,431,383,861		4,756,016,672	
% total area	100	110		90		30	

* Percentage area of peak after digestion; ** percentage area of peak after digestion and fermentation.

**Table 3 foods-14-03800-t003:** Total polyphenol content (TPC) and antioxidant capacity of *H. rhamnoides* before and after in vitro oro-gastroduodenal digestion and microbial fermentation with gut microbiota of healthy and constipated adults. TPC is expressed as mg gallic acid equivalents per g of extract (mg GAE/100 g); TEAC values from DPPH, ABTS, and FRAP assays are expressed as mmol Trolox equivalents per kg of extract (mmol TE/kg). Values are mean ± SD. Statistics: different letters indicate *p* < 0.05 for the digestion comparison; ns = not significant; symbol codes.

Assay	Before In Vitro Digestion	After In Vitro Digestion	After In Vitro Fermentation	
			Healthy Adults	Constipated Adults
FOLIN-CIOCALTEU	2924.2 ± 62.5	2235.7± 119.7	982.6 ± 19.9 **	587.3 ± 46.6
DPPH	201.8 ± 5.3	202.5 ± 6.6	30.6 ± 1.7 ^ns^	45.9 ± 4.9
ABTS	35.5 ± 2.7 ^a^	59.3 ± 0.7	21.2 ± 1.1 *	15.9 ± 0.6
FRAP	194.6 ± 30.8 ^a^	145.4 ± 3.8	2.9 ± 0 ****	0 ± 0

Values are expressed as the means ± standard deviation of biological replicates (*n* = 3). Each condition was assayed in technical triplicate and averaged. The TPC is expressed in mg of GAE/100 g of extract, while TEAC_DPPH_, TEAC_FRAP_, and TEAC_ABTS_ are expressed in mmol Trolox equivalents/kg of extract. Statistical significance in this comparison is indicated by different letters: ^a^
*p* < 0.05, ^ns^ non-significant, * *p* < 0.05, ** *p* < 0.01, **** *p* < 0.0001.

## Data Availability

The original contributions presented in the study are included in the article; further inquiries can be directed to the corresponding author.

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
