# Peer review of "In Vitro Influence of a Chemically Characterized *Hippophae rhamnoides* L. Fruit Extract on Healthy and Constipated Human Gut Microbiota Functionality and Aquaporin-3 Expression"

_foods, 2025, doi:10.3390/foods14213800_

Round 1

Reviewer 1 Report

Comments and Suggestions for Authors

In vitro effects of a chemically characterized Hippophae rhamnoides L. fruit extract on the expression of aquaporin-3 and bio-accessibility after simulated digestion and fermentation with microbiota isolated from healthy and constipated subjects

This manuscript presents a comprehensive in vitro investigation into the potential mechanisms of a Hippophae rhamnoides (sea buckthorn) fruit extract for alleviating functional constipation. Several major concerns must be addressed before the manuscript can be considered for publication.

  • Abstract and Introduction
  • The title of this paper is very long, making it difficult to understand.
  • In the abstract, the authors formulate long sentences, thus making it difficult to understand. Same for other sections.
  • Please add highlight sections.
  • Line 31, please, Specify the main or most abundant classes of compounds identified between 23 compounds.
  • Abstract lacks quantitative data that would significantly enhance its impact. What are the main SCFAs detected after fermentation? The authors should highlight this in the abstract. If possible, also the DPPH, ABTS, and FRAP values.
  • Consider adding a keyword related to the key analytical findings, such as short-chain fatty acids (SCFAs) or polyphenol bioaccessibility. This would greatly improve search ability of these keywords.
  • At the end of the introduction, after stating the main objective of this study, the authors should briefly summarize the results obtained.
  • Methods
  • Lines 93-94: Rephrase to: "Three batches of a commercial, dry, powdered hydroethanolic extract from H. rhamnoides fruits...".
  • Line 144: The concentration range for the extract alone is listed as "(2,1–210 ug/mL)". The comma as a decimal separator should be replaced with a period (2.1–210 µg/mL) for an international journal.
  • Lines 172-173: The sentence "The digestion process consisted of oral, gastric, and duodenal digestion processes" is redundant. It can be simplified.
  • In vitro fermentation section: The pooling of fecal samples from three donors per group is a common practice to average inter-individual variability. However, this choice must be justified, and its main limitation, the loss of information on individual variation in microbial response, should be acknowledged in the discussion.
  • DPPH results are expressed as ‘kg Trolox equivalent per gram’. This is incorrect; it should be ‘mg TE/g’ or, as used elsewhere, ‘mmol TE/kg’. A value in kilograms per gram is nonsensical. ABTS results are expressed as ‘mg Trolox equivalent per gram of sample’, while FRAP uses ‘mg TE/g’. The units must be standardized across all assays (all as mmol TE/kg or all as mg TE/g) for proper comparison and reporting in the results. This needs to be checked and corrected throughout the manuscript.
  • Lines 204-205: Why did the authors use 'hydrogen sulfide' in the growth medium and not “sodium sulfide”?
  • Specify that blot images will be provided in supplementary data.
  • Results
  • Lines 344-346: Values of 825.9 ± 199.6 mg GAE/g and 872.9 ± 465.9 mg GAE/g, it is right? These values are orders of magnitude higher than those reported in the literature. This must be corrected to mg GAE/100g or μg GAE/mg).
  • Lines 377-380: Healthy microbiota retained more ABTS+ activity" seems to contradict the presented data (21.2 and 15.9 mmol TE/kg), which actually shows the opposite. The healthy group had higher residual activity. The subsequent conclusion that constipated microbiota showed "greater use of radical scavengers" is the correct interpretation of the lower value. This sentence should be rephrased for clarity.
  • Figure 1, the authors are recommended to use the appropriate software to make this figure.
  • The authors should perform a correlation analysis between these 23 metabolites, antioxidants, SCFAs, and AQP-3 expression to better explain the mechanisms
  • Discussion
  • The discussion is very long; the authors must make an effort to be precise and succinct.
  • Line 550: This study is an in vitro investigation, yet the authors make strong clinical tests. This sentence: the extract 'helps restore SCFA levels in constipated subjects' and provides a 'protective effect.' These resonances are not supported by the data presented, as no clinical or in vivo experiments were performed. Such statements should be removed and replaced with conclusions that accurately reflect the experimental model.

Suggestion: "...suggests the extract has the potential to restore SCFA levels..." or "...may support optimal SCFA production."

  • Conclusion
  • For example, this long sentence, lines 617-623 should be rephrased for clarity.
  • The conclusion should be more succinct and should explicitly state the study's novel contribution, its limitations, and future research directions. Suggestions: The H. rhamnoides fruit extract is a rich source of bioactives that modulate gut microbiota function and host pathways relevant to constipation. Our in vitro findings demonstrate that the extract's bioaccessibility and antioxidant capacity are significantly altered by digestion and fermentation, with a notably higher degradation of polyphenols by constipated microbiota. Crucially, we identified two plausible mechanisms for its traditional use: the upregulation of the AQP-3 water channel, which may enhance stool hydration, and the promotion of a SCFA profile associated with healthy gut motility. Taken together, this study provides a mechanistic foundation for the use of sea buckthorn in constipation, underscoring the need for future clinical trials to confirm these effects in humans.

I recommend a Major Revision and look forward to reviewing an updated version that includes all necessary data and addresses the points raised above.

Reviewer 2 Report

Comments and Suggestions for Authors

Comments and suggestions

  1. The manuscript suffers from extensive textual overlap with previously published sources, as many paragraphs appear to be directly copied or only minimally modified. This raises concerns about plagiarism and originality. I strongly recommend that the authors thoroughly revise the text, ensure proper paraphrasing in their own words, and provide appropriate citations wherever prior work is being described.
  2. How does your UHPLC-Q-Orbitrap phenolic profiling advance beyond prior sea-buckthorn metabolite surveys; do you report any new constituents, structural confirmations (MS/MS, standards), or only tentative identifications already known in this species?
  3. The manuscript reports LC–MS analysis exclusively in negative ionization mode. Could the authors provide a rationale for this choice? For instance, was negative mode selected due to the predominance of phenolic acids and flavonoids in H. rhamnoides extracts, which generally ionize more efficiently as [M–H]⁻ ions? Please clarify whether positive ion mode was evaluated and excluded, or if negative mode alone was deemed sufficient for comprehensive metabolite coverage.
  4. Can you provide level-of-confidence assignments (e.g., MSI levels) and indicate which compounds were confirmed with authentic standards versus accurate-mass matches only, to substantiate novelty claims in Table 1? Do you match with MS2?
  5. line 118 reports the injection time in milliseconds, whereas line 121 reports it in seconds. Please revise to use a consistent unit of measurement throughout (either ms or s)
  6. The mass spec section state that an isolation window of 5 m/z was used. Could the authors explain the rationale for selecting this width? Was it chosen based on the instrument’s resolution setting, the need to balance sensitivity and selectivity, or prior optimization for these analytes?
  7. mass accuracy was controlled within a tolerance of 5 ppm for analyte identification. Could the authors clarify how this accuracy was ensured, did they apply an internal lock mass calibration during acquisition, or was external calibration performed prior to analysis?
  8. The authors used the Folin–Ciocalteu assay to determine total phenolic content. Could the authors clarify why this method was chosen? Since the Folin–Ciocalteu reagent reacts not only with phenolics but also with other reducing compounds ( ascorbic acid, sugars, ), how do you justify its specificity for phenolics in this extract.
  9. In the antioxidant assays, DPPH preparation is described using dry weight (mg) of DPPH, whereas ABTS is reported in millimolar units. Please make the units consistent across assays, or provide a clear rationale for reporting them differently.
  10. The Methods state triplicate digestions and fermentations with pooled donors, yet pooling three fecal donors per arm reduces the number of independent biological units and challenges the validity of parametric tests that assume independence; please clarify what constituted the statistical “n” for SCFA and antioxidant analyses and whether the triplicates were technical or biological replicates. 
  11. Only 23 compounds were reported despite Orbitrap HRMS; please clarify whether co-elution limited identifications and how you assessed/managed it.
  12. Compound identification is missing in method section
  13. Please clarify in the Methods that ‘reduction to 15% A’ indicates increasing solvent B (methanol) to strengthen elution, and that the final 2 min at 100% A represent the washing/re-equilibration phase. This would improve clarity for readers less familiar with LC gradient terminology.
  14. Since static INFOGEST digestion is now standard, what is conceptually new in your application here—parameterization, matrix-specific insights, or a benchmark comparison against other constipation-relevant botanicals?
  15. Pooling three donors per arm may mask inter-individual variance; can you justify pooling with variance estimates or sensitivity analyses, and explain how this affects claims about differential responses in healthy versus constipated microbiota?
  1. SCFA shifts after polyphenol fermentation are well documented; what is the specific incremental insight here for functional constipation (e.g., effect sizes relative to baseline or to positive controls) that would not be anticipated from existing literature?
  2. AQP-3 modulation has been shown for several phytochemicals; what makes the AQP-3 upregulation by sea-buckthorn extract mechanistically distinct (e.g., active constituents traced to effect, receptor or signaling axis), rather than another    instance of a known phenomenon?
  1. Are the in-vitro concentrations that upregulate AQP-3 achievable in the human colon lumen or mucosa after realistic oral dosing and first-pass metabolism; can you supply exposure estimates or physiologically based reasoning to support translational plausibility?
  2. Polyphenols are known to undergo extensive first-pass metabolism and conjugation , at liver, (e.g., glucuronidation, sulfation). How do the authors expect these metabolites to influence the observed AQP-3 effects or SCFA production?
  3. The in vitro models do not account for systemic absorption and tissue distribution. Can the authors discuss whether the active compounds identified would reach colonic epithelial targets in vivo, or whether microbial metabolites might be the true effectors?
  4. Have the authors evaluated potential cytotoxicity at the concentrations used in cell assays, and can they comment on any known safety limitations of sea buckthorn extracts in long-term or high-dose consumption?
  5. Which identified compounds (or fractions) track with the AQP-3 effect across dose or fractionation; can you narrow the mechanism to defined constituents (e.g., isorhamnetin derivatives, caffeic acid family) rather than attributing it to the unfractionated mixture?
  6. How does sea-buckthorn’s integrated profile (polyphenol stability, SCFA increase, AQP-3 modulation) compare head-to-head with a recognized positive control for constipation (e.g., inulin, lactulose) in your models, and does this change the strength of novelty claims?
  7. Given ongoing and registered clinical interest in sea-buckthorn for bowel function, what new knowledge from your workflow would change clinical development decisions (e.g., biomarker guidance, responder stratification by microbiota, or dose/formulation optimization)?
  8. The current figure and table captions are too brief. Please revise them so that each caption is self-contained and explains all elements. For figures, include clear descriptions of the x- and y-axes, units, sample groups, and statistical annotations (letters, asterisks, error bars). For tables, explain every column heading, including abbreviations, units of measurement, and statistical notation. Captions should allow readers to fully interpret the content without referring back to the main text.
  9. The peak area should be presented in proper scientific notation where appropriate
  10. Please shorten your topic and conclusion so that readers can easily understand your work.
  1. Your study investigates compounds after simulated digestion. Since the digestion process involves enzymatic hydrolysis that breaks proteins into peptides, why did you not also analyze the peptide fraction? Including or at least discussing this component would strengthen the work, as peptides may contribute to the observed biological effects.
  2. Please shorten long sentences, unify terminology, check consistency of units and abbreviations

Reviewer 3 Report

Comments and Suggestions for Authors

This study investigated the effects of hydroethanolic extract of Hippophae rhamnoides fruit on intestinal function. Metabolite profiling, along with simulated in vitro oro-gastrointestinal digestion and fecal microbiota fermentation from healthy and constipated adults, was performed using UHPLC Q-Orbitrap HRMS to determine polyphenol metabolites. In addition, the extract was found to upregulate AQP-3 expression in HT-29 cells.

The following suggestions are offered for revision.

Line 97

What is the fixed standardization?

Line 117

The full name of FWHM should be described.

Line 144

2.1–210

Line 309

It should be metabolite profile rather than metabolic profile.

Line 318 (Table 1)

(1)Please revise the title of Table 1.

“Table 1. Tentatively identified compounds and their corresponding analytical parameters (n=23).”

(2) 17: syringic acid

Line 335

apigenin aglycone

Line 345

It should be 9825.9 ± 199.6 mg GAE/g.

Line 346

It should be 5872.9 ± 465.9 mg GAE/g.

Line 350 (Table 2)

syringic acid

Line 370 and Table 3

DPPH (before in vitro digestion) 201.8 ± 5.3

Line 384 and Table 3

FRAP 2.9 ± 0

Line 402~403

“SCFAs produced in response to H. rhamnoides fruit extract in the healthy group (56.02 mM) compared to the constipated group (48.57 mM).”

The SCFA analysis was carried out in triplicate for each experimental condition. The data should be presented as means ± standard deviation.

Line 432 (figure 2)

(1)2.1, but not 2,1.

(2) What is the control (ctrl)?

Line 433

2.1–210

Reviewer 4 Report

Comments and Suggestions for Authors

Hi, here are some observations.

Title

It is recommended to improve the wording and structure of the article title. It is too long.

Abstract

Review the journal's guidelines. Check if abbreviations are allowed in the abstract and correct any misspelled ones: DPPH, ABTS.

Justify the abstract text.

Manuscript

Review all references; some are not spelled correctly according to the established formats.

DPPH, ABTS. should be written throughout the manuscript.

Figure 1 needs to be improved, adding names and units on the Y axis.

The quality of the figures needs to be improved.

In the section of  references some or several require attention according to the format established by the journal. Review and address them.

It is important that the description of extracts and reported values ​​of DPPH, ABTS, and FRAP are not described in the results and conclusions section, explaining why the three techniques were selected, what advantages, and what results can be discussed, mentioned, or highlighted after applying the aforementioned techniques.

Round 2

Reviewer 1 Report

Comments and Suggestions for Authors

After a thorough check, the authors have greatly improved the quality of the manuscript by taking into account all my comments and suggestions.

Author Response

We would like to thank Reviewer 1 for his/her suggestion. The paper was evaluated by a native speaker 

Reviewer 2 Report

Comments and Suggestions for Authors

The reviewer response is largely coherent and scientifically defensible, but several promised edits are only partially reflected in the revised manuscript

The manuscript still contains internal unit inconsistencies in the MS-acquisition paragraph despite the response letter’s assurance that milliseconds would be used throughout; specifically, maximum injection time appears once as “0.2 s” for full MS and again as “200 ms” for AIF within the same section, which contradicts the author’s stated correction and risks confusion for method replication, so standardizing to milliseconds across that paragraph is necessary to align text with the response.

The authors explain that mass accuracy within ±5 ppm was ensured by routine external calibration before each analytical batch, yet the Methods do not explicitly state this calibration practice and only mention the tolerance itself; adding some sentence documenting external calibration will reconcile the manuscript with the response and strengthen the methodological record for readers.

The justification for using Folin–Ciocalteu—acknowledging non-phenolic reducing-agent interference and emphasizing cross-study comparability—is presented in the response but is absent from the Methods section that currently lists only the assay steps and units; inserting a brief rationale after the TPC protocol would honor the response’s commitment and preempt renewed reviewer concern about specificity.

Assay-unit harmonization remains incomplete even though the response letter states that outcomes for DPPH, ABTS, and FRAP are uniformly reported as mmol Trolox equivalents per kg; in the FRAP subsection, one sentence still reports “mg Trolox equivalent/g” immediately followed by “mmol TE/kg,” which should be edited to a single unit to match the declared policy and avoid ambiguity in cross-assay comparisons.

Several promised clarifications appear correctly implemented and could be retained as resolved, including the LC gradient explanation (15% A equals increased methanol, and the 100% A step is the wash/re-equilibration) and the rationale for negative-mode selection with MSI-level reporting, but these do not mitigate the need to fix the remaining calibration and unit-consistency gaps flagged above.

Concise the topic.

Author Response

The reviewer response is largely coherent and scientifically defensible, but several promised edits are only partially reflected in the revised manuscript.

1. The manuscript still contains internal unit inconsistencies in the MS-acquisition paragraph despite the response letter’s assurance that milliseconds would be used throughout; specifically, maximum injection time appears once as “0.2 s” for full MS and again as “200 ms” for AIF within the same section, which contradicts the author’s stated correction and risks confusion for method replication, so standardizing to milliseconds across that paragraph is necessary to align text with the response.

Answer: We thank the reviewer for spotting this inconsistency. We indeed stated in our previous response that MS-acquisition parameters would be expressed uniformly in milliseconds. In the revised version we have now standardized the two occurrences of maximum injection time in the UHPLC–HRMS section: the value previously reported as “0.2 s” for the full MS experiment has been corrected to “200 ms,” so that it matches the “200 ms” already reported for AIF. We have also made explicit that the scan duration for AIF is “100 ms.” This way, the whole paragraph now uses a single unit (ms) for time-related acquisition parameters.

2. The authors explain that mass accuracy within ±5 ppm was ensured by routine external calibration before each analytical batch, yet the Methods do not explicitly state this calibration practice and only mention the tolerance itself; adding some sentence documenting external calibration will reconcile the manuscript with the response and strengthen the methodological record for readers.

Answer: We thank the reviewer for the comment. As suggested, we have added a sentence in the Methods section specifying that routine external calibration was performed before each analytical batch to ensure mass accuracy within ±5 ppm.

3. The justification for using Folin–Ciocalteu, acknowledging non-phenolic reducing-agent interference and emphasizing cross-study comparability, is presented in the response but is absent from the Methods section that currently lists only the assay steps and units; inserting a brief rationale after the TPC protocol would honor the response’s commitment and preempt renewed reviewer concern about specificity.

Answer: We thank the reviewer for this suggestion. In the revised manuscript, immediately after the TPC protocol (Section 2.6.1), we have added a sentence explicitly acknowledging that Folin–Ciocalteu is not fully specific for phenolic compounds and may detect other reducing agents, and that we nonetheless selected it to ensure comparability with previous studies on similar matrices. This addition aligns the manuscript text with the explanation given in our response and should prevent renewed concern about method specificity.

4. Assay-unit harmonization remains incomplete even though the response letter states that outcomes for DPPH, ABTS, and FRAP are uniformly reported as mmol Trolox equivalents per kg; in the FRAP subsection, one sentence still reports “mg Trolox equivalent/g” immediately followed by “mmol TE/kg,” which should be edited to a single unit to match the declared policy and avoid ambiguity in cross-assay comparisons.

Answer: We appreciate the reviewer’s careful reading. The mixed units in the FRAP subsection (“mg Trolox equivalent/g” followed by “mmol TE/kg”) were an oversight during revision. Specifically, the residual “mg TE/g” instance in the FRAP subsection has been corrected to mmol TE/kg. This harmonization removes ambiguity.

5. Several promised clarifications appear correctly implemented and could be retained as resolved, including the LC gradient explanation (15% A equals increased methanol, and the 100% A step is the wash/re-equilibration) and the rationale for negative-mode selection with MSI-level reporting, but these do not mitigate the need to fix the remaining calibration and unit-consistency gaps flagged above.  

Concise the topic.

Answer: We appreciate the reviewer’s positive feedback. The requested corrections regarding unit consistency and calibration have been implemented. The measurement units have been revised for consistency throughout the manuscript, and a description of the external calibration procedure has been added to the Materials and Methods section.